# Monochromatic multicomponent fluorescence sedimentation velocity for the study of high-affinity protein interactions

Huaying Zhao[1], Yan Fu[2], Carla Glasser[3], Eric J Andrade Alba[2], Mark L Mayer[3], George Patterson[2], Peter Schuck[1]*

[1]Dynamics of Macromolecular Assembly Section, Laboratory of Cellular Imaging and Macromolecular Biophysics, National Institute of Biomedical Imaging and Bioengineering, National Institutes of Health, Bethesda, United States; [2]Section on Biophotonics, National Institute of Biomedical Imaging and Bioengineering, National Institutes of Health, Bethesda, United States; [3]Laboratory of Cellular and Molecular Neurophysiology, Porter Neuroscience Research Center, National Institute of Child Health and Human Development, National Institutes of Health, Bethesda, United States

*For correspondence: schuckp@mail.nih.gov

Competing interests: The authors declare that no competing interests exist.

**Abstract** The dynamic assembly of multi-protein complexes underlies fundamental processes in cell biology. A mechanistic understanding of assemblies requires accurate measurement of their stoichiometry, affinity and cooperativity, and frequently consideration of multiple co-existing complexes. Sedimentation velocity analytical ultracentrifugation equipped with fluorescence detection (FDS-SV) allows the characterization of protein complexes free in solution with high size resolution, at concentrations in the nanomolar and picomolar range. Here, we extend the capabilities of FDS-SV with a single excitation wavelength from single-component to multi-component detection using photoswitchable fluorescent proteins (psFPs). We exploit their characteristic quantum yield of photo-switching to imprint spatio-temporal modulations onto the sedimentation signal that reveal different psFP-tagged protein components in the mixture. This novel approach facilitates studies of heterogeneous multi-protein complexes at orders of magnitude lower concentrations and for higher-affinity systems than previously possible. Using this technique we studied high-affinity interactions between the amino-terminal domains of GluA2 and GluA3 AMPA receptors.

## Introduction

The dynamic formation of multi-protein complexes is a key step in the assembly of supramolecular structures and in the regulation of many cellular processes (*Wu, 2013*; *Li et al., 2012*; *Gavin et al., 2002*; *Krogan et al., 2006*; *Wu and Fuxreiter, 2016*). For example, in immunological signal transduction the assembly of adaptor protein complexes into micro-clusters after T-cell activation plays a critical role in the sensitivity and specificity of activation (*Sherman et al., 2011*; *Dustin and Groves, 2012*). Another well-known multi-protein complex is the post-synaptic density, a large structure assembled *via* interactions between many different scaffolding proteins, signaling proteins and ligand gated ion channels, that regulates postsynaptic neurotransmission and plasticity (*Kennedy, 2000*; *Ferré et al., 2007*; *Kumar and Mayer, 2012*). Many of the protein interactions involved in the assembly of such molecular machinery are multivalent (*Li et al., 2012*; *Houtman et al., 2006*;

**eLife digest** Many proteins in cells combine to form molecular machines or complexes that carry out specific processes inside cells. Analytical ultracentrifugation is a technique commonly used to explore the physical properties of proteins and their complexes and in this way to gain insights into the biological roles of these molecules. The technique involves spinning a sample containing the molecules to generate a strong centrifugal force, while monitoring the movement of the molecules. Under these conditions, molecules with different sizes and masses sink – or "sediment" – at different rates, so individual proteins and their complexes can be clearly distinguished.

Analytical ultracentrifugation was recently extended to make it possible to detect fluorescent tags added on to proteins. This advance allowed researchers to study more dilute samples or complexes that are held together especially tightly. However, only tags of a single color can be detected because of physical constraints of the fluorescent detection system. This meant that only one kind of fluorescent signal could be tracked at any one time. However, a group of fluorescent tags called photoswitchable fluorescent proteins (psFPs) offer new opportunities for detecting multiple signals. This is because these psFPs switch between fluorescent and non-fluorescent states while being detected in the ultracentrifuge.

Zhao et al. have now exploited this unique photoswitching property by accurately measuring how fast a number of psFPs switched between fluorescent and non-fluorescent states while they were sedimenting. Each different psFPs switched in a distinct way, even for psFPs of the same color, meaning that each psFP could be identified from its switching rate, similar to identifying a person from their fingerprints. This discovery allowed Zhao et al. to distinguish different psFPs in a mixed sample as if they had different colors.

Further experiments went on to demonstrate that this approach could identify the binding proteins in a protein mixture made of three components, and be used to study a biologically important protein complex that can itself exist in two distinct forms. The approach will therefore provide a valuable tool to observe different components in a complex individually and will provide researchers the opportunity to study how mixed protein complexes form at very low concentrations. Future developments of the approach may make it possible to study other properties of protein complexes such as their overall shape and their behavior under conditions that mimic those inside the cell.

*Coussens et al., 2013*). This often allows structurally polymorph complexes to co-exist (*Wu and Fuxreiter, 2016*), posing formidable challenges for any biophysical method to elucidate basic architectural principles and driving forces, which requires the study of reversible interactions of multiple protein components with multiple states.

Sedimentation velocity analytical ultracentrifugation (SV) is a classical technique that allows determination of the number, size, and shape of reversibly formed protein complexes, and provides information on their affinity, stoichiometry and binding kinetics (*Schuck, 2013*, *2015*). Though a long established technique, it is worth recapitulating the basic principles of SV (*Figure 1*). In SV the spatio-temporal evolution of macromolecular concentration profiles in a sample solution after application of a strong centrifugal field is optically monitored in real-time. SV has unique opportunities for studying protein interactions, since—different from separation techniques—faster sedimenting protein complexes will always remain in a bath of slower-sedimenting constituent components, such that the association/dissociation of non-covalent complexes is maintained throughout the experiment (*Figure 1*). Since sedimentation takes place free in solution, the analysis can be based on first principles and mathematical models for the sedimentation/diffusion process, and modern size-distribution analysis results in sedimentation coefficient distributions with high hydrodynamic resolution. Thus, SV has emerged as a powerful technique in the study of the solution state behavior of complex interacting systems of macromolecules, including ion-channels, adaptor proteins, membrane proteins, nucleic acids, and carbohydrates (*Kumar and Mayer, 2012*; *Houtman et al., 2005*; *le Maire et al., 2008 Niewiarowski et al., 2010*; *Padrick and Brautigam, 2011*; *Harding et al., 2015*; *Jose et al., 2012*). Extended to multi-signal analysis SV can distinguish different sedimenting

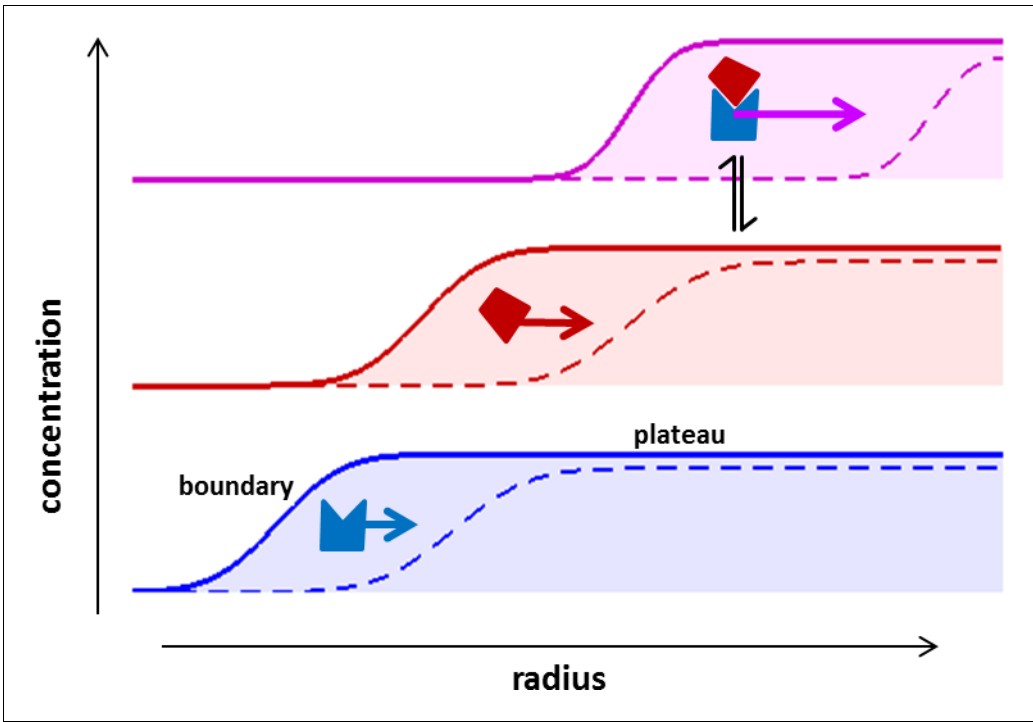

**Figure 1.** Concentration profiles in a sedimentation velocity experiment. Two different macromolecular components are depicted (blue and red) reversibly forming a complex (magenta). As a result of centrifugal force at 200,000–300,000 g, macromolecules sediment at a rate determined by their mass, density, and Stokes radius (or translational friction coefficient) (*Svedberg and Rinde, 1924*). The velocity of sedimentation normalized relative to the centrifugal field strength is expressed in the molecular sedimentation coefficient *s*. With time, transport clears the region of the solution column closest to the center of rotation and a moving front is formed – the sedimentation boundary – that separates the cleared zone from a zone of constant concentration named the solution plateau region. While the boundary moves with time (dashed vs solid line), the concentration in the plateau region continuously decreases, solely due the radial geometry of sedimentation resulting in an increase in intermolecular distances (for a detailed description, see *Schuck et al., 2015*). If protein interactions cause complexes to form, these generally sediment faster and therefore migrate through a bath of slower sedimenting free constituent components. This allows association/dissociation reactions to continuously occur in a way that reflects equilibrium and kinetic properties of the interaction, at the same time as the complex boundaries are hydrodynamically resolved (*Schuck, 2010*). The temporal evolution of the boundary shapes is governed by macromolecular diffusion and polydispersity, and the latter can be extracted by mathematical modeling of experimental data in form of sedimentation coefficient distributions (*Schuck, 2000*).

components from characteristic extinction properties, directly revealing binding stoichiometries and resolving co-existing complexes (*Houtman et al., 2006*; *Coussens et al., 2013*; *Padrick and Brautigam, 2011*; *Balbo et al., 2005*; *Barda-Saad et al., 2010*). For example, through applications of this approach an essential mechanism for the formation of signaling particles in T-cell activation (*Houtman et al., 2006*; *Coussens et al., 2013*; *Barda-Saad et al., 2010*) was discovered to be the multivalent intracellular oligomerization of LAT via three-component adaptor protein complexes (*Houtman et al., 2006*). Such biophysical multi-protein solution studies naturally complement superresolution fluorescence imaging and co-localization studies of live cells (*Sherman et al., 2011*; *Houtman et al., 2006*; *Coussens et al., 2013*; *Barda-Saad et al., 2010*). But, unfortunately, traditional SV is limited in several ways by optical detection systems that generally require the use of micromolar concentrations of purified proteins.

Recently, analytical ultracentrifugation was enhanced by the availability of a commercial fluorescence optical detection system (FDS), that uses confocal detection radially scanning the sample in the spinning rotor (*MacGregor et al., 2004*) (*Figure 2a*). After accounting for characteristic data features, the FDS allows highly quantitative analyses of the sedimentation process (*Zhao et al., 2013b*),

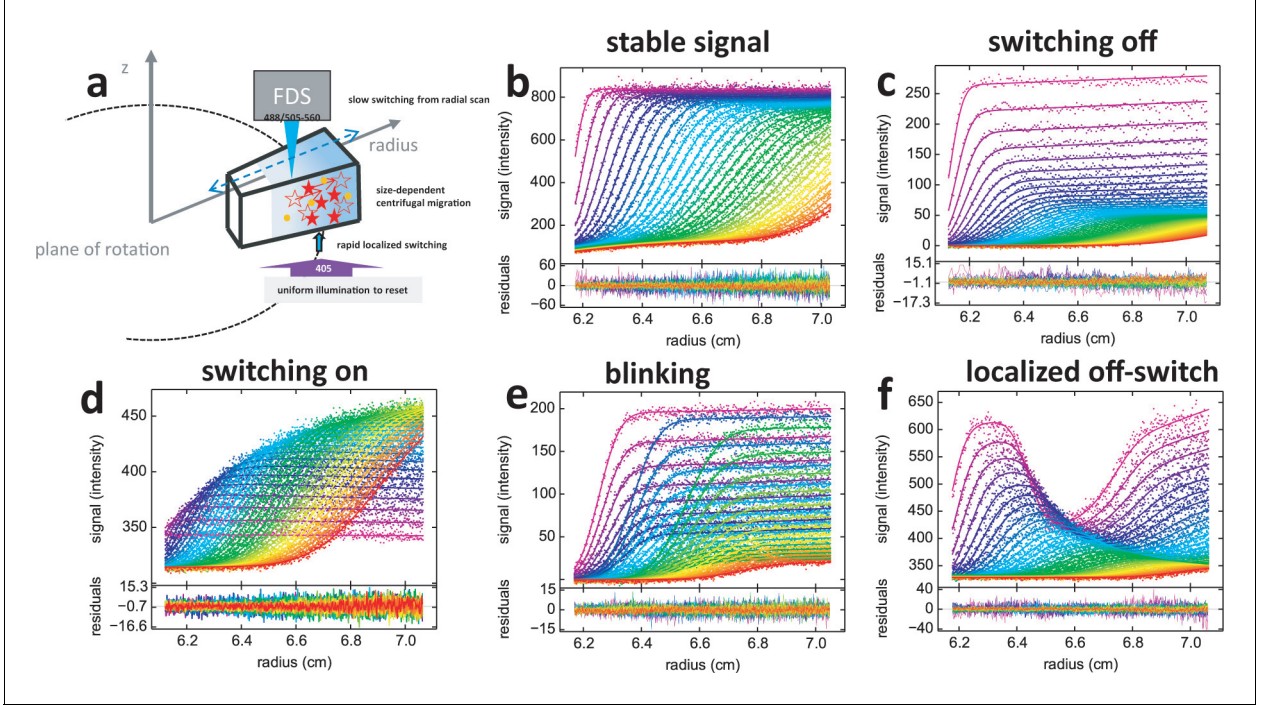

**Figure 2.** Principle of fluorescence detected sedimentation velocity and optical switching of psFPs. (a) Schematic setup: A rotating sample solution is scanned radially in a confocal configuration with 488 nm excitation (13 mW unless mentioned otherwise), inducing slow photoswitching of psFPs. Centrifugal forces cause strongly size-dependent migration, as depicted in *Figure 1*. Optionally, localized exposure at 488 nm or uniform illumination at 405 nm can further modulate the spatio-temporal signal. (b-f) Radial fluorescence scans (dots, color indicating times in order purple-blue-green-yellow-red; every 2nd scan shown) during sedimentation at 50,000 rpm and 20°C for different fluorophores and illumination conditions. More detailed inspection of the data is possible from the associated movies. Solid lines are the best-fit with a single-species (c to f) or distribution (b) model for the sedimentation/diffusion/photoswitching process; residuals are shown in the lower panels. (b) For DL488-GluA2 (5 nM) only a small depletion of plateau signal with time occurs, due to sample dilution as geometrically predicted from radial migration in the sector-shaped sample solution. (see *Video 1*) (c) rsEGFP (30 nM) exhibits an exponential depletion of the sedimentation signal (see *Video 2*). (d) Exposure in the scanning beam causes Padron (20 nM) to switch from predominantly dark to a fluorescent state, causing an exponentially saturating signal increase with time (see *Video 3*). (e) The sedimentation of 10 nM rsEGFP2-GluA3 is recorded with a 25 mW scanning beam, interrupted by 120 s exposures with 405 nm light at time points 40 min (prior to the purple scans), 64 min (prior to the blue scans), and 102 min (prior to the green scans), each time switching fluorophores from dark state back to the fluorescent state (see *Video 4*). (f) 5 min into the sedimentation run of 5 nM rsEGFP2-GluA3, a localized initial trough was generated by holding the scanning beam stationary at 6.5 cm for 20 min, locally causing strong conversion of fluorophores into the dark state. Standard scans of the sedimentation process follow, highlighting diffusion into the trough superimposed by migration and slow switching off (see *Video 5*).

extending the sensitivity of SV by several orders of magnitude into the low picomolar range (*Zhao et al., 2014a*; *Le Roy et al., 2015*). This enables the measurement of binding energies in self- and hetero-association with very high affinity (*Zhao et al., 2012*; *2013a*; *2014a*), and also makes it possible to study proteins that are relatively scarce, not well purified, and in some cases even in cell extracts (*Le Roy et al., 2015*; *Polling et al., 2013*; *Kingsbury and Laue, 2011*; *Kokona et al., 2015*). Unfortunately, these advantages come with a drawback of having only a single excitation wavelength, either 488 nm or 561 nm, due to the technical constraints of accommodating a move-able confocal optical system inside the evacuated rotor chamber of the ultracentrifuge. Thus, spectral discrimination of multiple components is not available in AUC fluorescence detection, which significantly limits the study of multi-protein complexes.

In the present work, we embark on a different approach for multi-component analysis based on photophysical properties of photoswitchable fluorescent proteins (psFPs). The psFPs make up a class of fluorescent proteins that can be actively switched between fluorescent and non-fluorescent states using different wavelengths of illumination. While they have been engineered for entirely different purposes in nanoscience and fluorescence imaging, we have previously observed that under the illu-mination conditions of FDS-SV they are induced to slowly switch by virtue of the excitation light

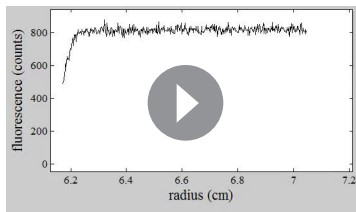

**Video 1.** Example of conventional boundaries from molecules with stable signal. Time-course of signal profiles for DL488-GluA2 sedimenting at 50,000 rpm, as shown in *Figure 2b*.

when radially scanning the spinning sample (*Zhao et al., 2014b*). Even though the mechanism of photoswitching in psFPs generally may involve multiple states, in the low-power exposure that occurs during sedimentation using the FDS the process is quantitatively modeled very well as a single exponential. This is exploited and further developed in the present work. Different classes of psFPs exhibit different time-courses of photoswitching in FDS-SV, and may be switched on to a fluorescent state or switched off to a dark state. We show that this process is highly quantitative, and how this signal change can be manipulated spatially and temporally during sedimentation. The new spatio-temporal signal dimension is folded into the computational analysis of the sedimentation process, and thus offers an avenue for monochromatic multi-component (MCMC) detection. Using existing commercial FDS instrumentation, the MCMC approach allows us to simultaneously determine separate sedimentation coefficient distributions for each class of fluorophore, which can be used to determine the identity and binding stoichiometry of hydrodynamically resolved complexes of psFP-tagged proteins.

We first demonstrate experimental proof of principle of exploiting the photo-switching kinetics as a novel aspect of fluorophore analysis. Using psFPs commonly employed in super-resolution microscopy, we show an example for the identification of binding partners in a three-component protein mixture. We then use this approach to study the high-affinity interactions of glutamate receptor GluA2 and GluA3 amino terminal domains, which engage in competitive homo-dimerization and hetero-dimerization processes that are thought to control the combination of receptor subtypes into diverse homomeric and heteromeric ion channel tetramers with different gating properties (*Kumar and Mayer, 2012*; *Rossmann et al., 2011*; *Herguedas et al., 2016*).

## Results

### Fluorescence signals of psFPs in sedimentation velocity

Fluorescence SV signals of psFPs are highly unusual when compared to the temporal evolution of concentration profiles in traditional AUC (*Figure 2b*), in that they exhibit sedimentation boundaries modulated by characteristic signal magnification or diminution on the time-scale of sedimentation (*Figure 2c,d*). This is caused by the 488 nm excitation beam of the FDS scanner which induces photoswitching between fluorescent and dark states. But in contrast to typical power densities of ~ kW/cm$^2$ for photoswitching on the millisecond time-scale (*Grotjohann et al., 2012*), the transient exposure in FDS-SV during radial scanning and sample rotation leads to a time-averaged incident power density that is ~10$^5$-fold weaker. This slows the photoswitching kinetics down to the time-scale of

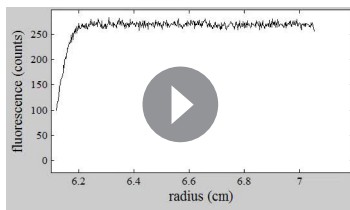

**Video 2.** Example of decreasing boundaries of molecules switching off with 488 nm exposure. Time-course of signal profiles for rsEGFP sedimenting at 50,000 rpm while undergoing continuous slow signal depletion through the 488 nm illumination of the FDS scanner, as shown in *Figure 2c*.

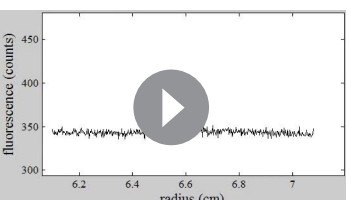

**Video 3.** Example of growing boundaries of molecules switching on with 488 nm exposure. Time-course of signal profiles for Padron sedimenting at 50,000 rpm while undergoing continuous slow signal amplification through the 488 nm illumination of the FDS scanner, as shown in *Figure 2d*.

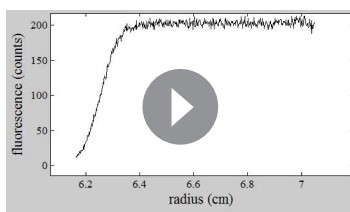

**Video 4.** Example of blinking boundaries. Time-course of signal profiles for rsEGFP2-GluA3 sedimenting at 50,000 rpm while undergoing continuous slow signal depletion through the 488 nm illumination of the FDS scanner, in combination with periodic signal reset through 2 min. pulses of strong 405 nm illumination. (See also *Figure 2e*.)

hours, commensurate with the sedimentation process in SV. A prerequisite for exploiting this new temporal dimension for the multi-component decomposition of fluorescence SV data is the ability to precisely describe the signal evolution of the individual psFPs. We have developed a model *Equation 4* that assumes a single-step process with constant quantum efficiency for switching, while accounting for the radially non-uniform exposure during scanning (caused by psFPs transitioning through the beam in a shorter time as they migrate to higher radii at the same angular velocity). Combined with a description of molecular sedimentation and diffusion, the model predicts signal boundaries to be subject to an exponential overall signal modulation with slightly radially sloping solution plateaus, with a small positive slope for those switching off (*Figure 2c*) and a small negative slope for FPs switching on (*Figure 2d*) under 488 nm illumination. As shown in the examples of *Figure 2*, fits of this model to within the noise of data acquisition can routinely be achieved for the fluorescence sedimentation data of diverse FPs, including, for example, the rapidly de-activating rsEGFP (*Grotjohann et al., 2012*) and the strongly activating Padron (*Andresen et al., 2008*) (*Figure 2c,d*). From the fits of the single component samples, we can obtain the relative amplitude and time-constant of photoswitching for different fluorophores, which serves as a highly reproducible, characteristic temporal tag. For example, for rsEGFP with a scanning beam of 13 mW we measure a depletion rate of 5.08 [4.97–5.19; 95% CI]$\times 10^{-4}$/sec, approaching a final fluorescence of 13.4 [13.0–13.9; 95% CI]% its initial value, associated with a particle sedimentation coefficient of 2.50 [2.45–2.54; 95% CI] S and apparent molar mass of 29.5 [26.5–33.0; 95% CI] kDa. By contrast, as previously established (*Zhao et al., 2014b*), no photophysical processes are detectable under this illumination for other fluorophores such as fluorescein-based DL488 (*Figure 2b*) and standard EGFP.

## MCMC decomposition of mixtures

The strikingly different signal patterns in sedimentation of psFPs can be utilized for the computational decomposition of sedimentation data of mixtures into separate sedimentation coefficient distributions for each differently tagged component. As an initial proof of principle, *Figure 3a* shows sedimentation data of a mixture of rsEGFP2 and FITC-BSA. The shape of the signal boundaries is governed jointly by the signal time-domain, polydispersity in the sedimentation coefficient distribution, as well as diffusion. The initially strongly decreasing plateau intensity over time is a characteristics of rsEGFP2 signals that can be readily visually discerned and distinguished from the stable fluorescein signal that causes the plateaus to stabilize at approximately half the total loading signal. In addition to the decreasing plateau, the decreasing amplitude of the sedimentation boundary also carries significant information on the time-dependent rsEGFP2 signal. This is highlighted in *Figure 3b* by the failure of an impostor fit with conventional boundary analysis, in which the decreasing plateau level are compensated for with *ad hoc* inclusion of time-dependent baseline offsets (usually absent in FDS-SV [*Schuck et al., 2015*; *Zhao et al., 2013b*]).

In the MCMC analysis the fluorophore photoswitching parameters were fixed to predetermined values of the individual fluorophores (run side-by-side in a different rotor position). The

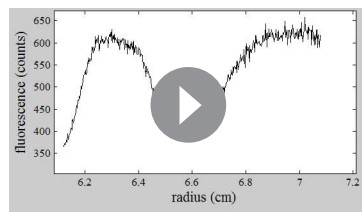

**Video 5.** Example of FRAP-like sedimentation. Initial localized illumination with the stationary 488 nm excitation beam of the FDS creates a trough in the signal of rsEGFP2-GluA3. The sedimentation/diffusion process at 50,000 rpm causes the relative trough to diminish, a process that is superimposed by the standard overall signal depletion from the 488 nm scanner. (See also *Figure 2f*).

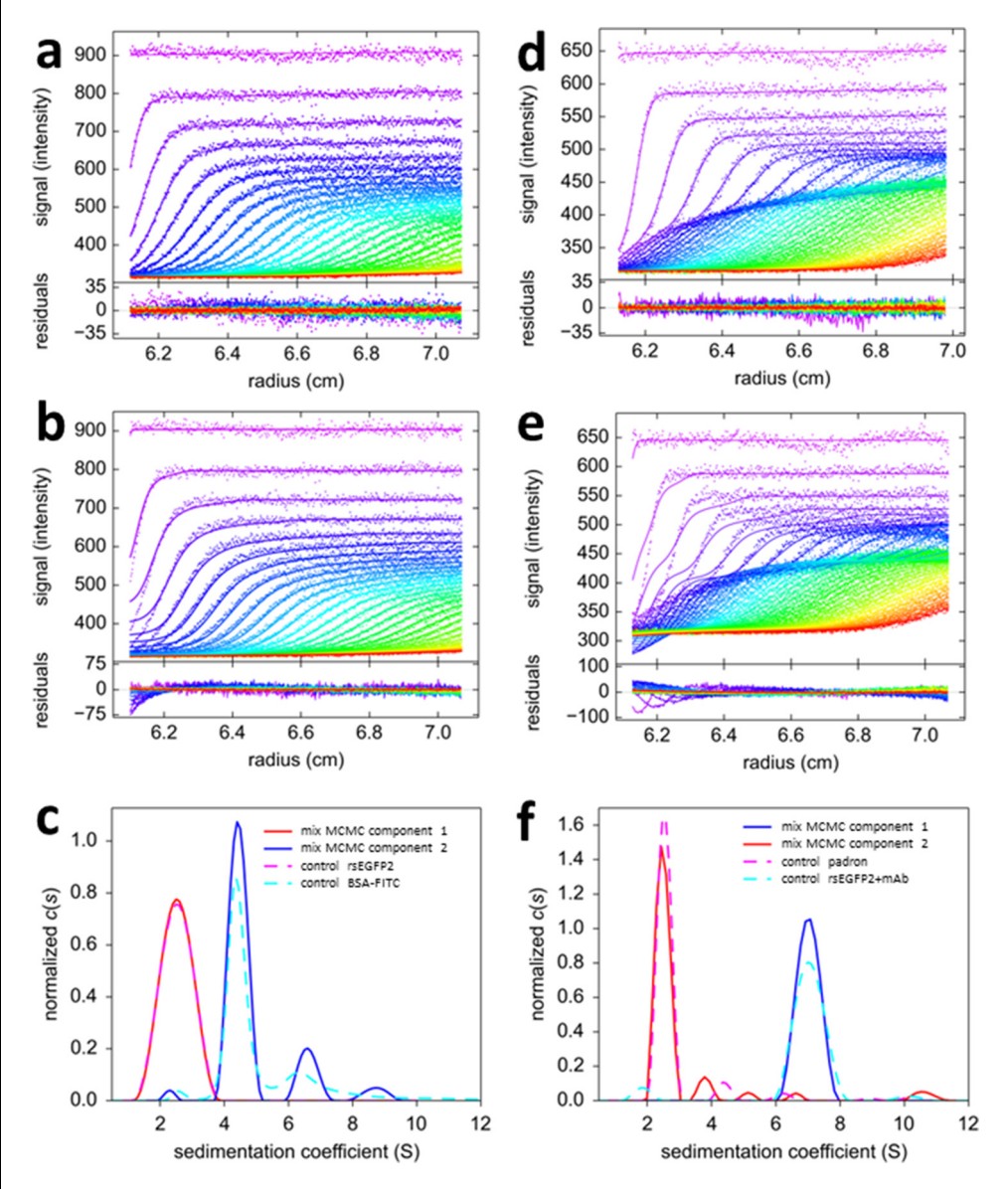

**Figure 3.** Mono-chromatic multi-component (MCMC) decomposition of mixtures. (**a**) Evolution of radial fluorescence profiles of 20 nM rsEGFP2 and 30 nM FITC-BSA sedimenting at 50,000 rpm and 20°C. Raw data are shown as points with higher color temperature indicating the passage of time (only every second scan shown), and solid lines are the best-fit based on the MCMC distribution model with two components, producing residuals as shown in the lower panel attached. (**b**) An impostor fit with a conventional *c(s)* analysis not accounting for time-dependent signal increments, but compensating their effect on plateau levels by artificial inclusion of time-dependent baseline offsets, creates large misfit in the boundary region. (**c**) The resulting sedimentation coefficient distributions of each fluorophore component from the MCMC analysis of the mixture (solid lines) and in separate control experiments with individual samples (dashed lines). (**d**) Fluorescence sedimentation data acquired under the same conditions for a mixture of 20 nM rsEGFP2, 20 nM Padron, and 50 nM anti-GFP mAb, presented in the same format as (**a**). (**e**) Analogous to (**b**), showing the best-fit 'conventional' boundary model not accounting for time-dependent signal increments while including artificial baseline parameters. (**f**) Resulting sedimentation coefficient distributions from the MCMC analysis (solid lines) and standard control experiments (dashed lines).

The following figure supplement is available for figure 3:

**Figure supplement 1.** Sedimentation signals of rsEGFP2-GluA3.

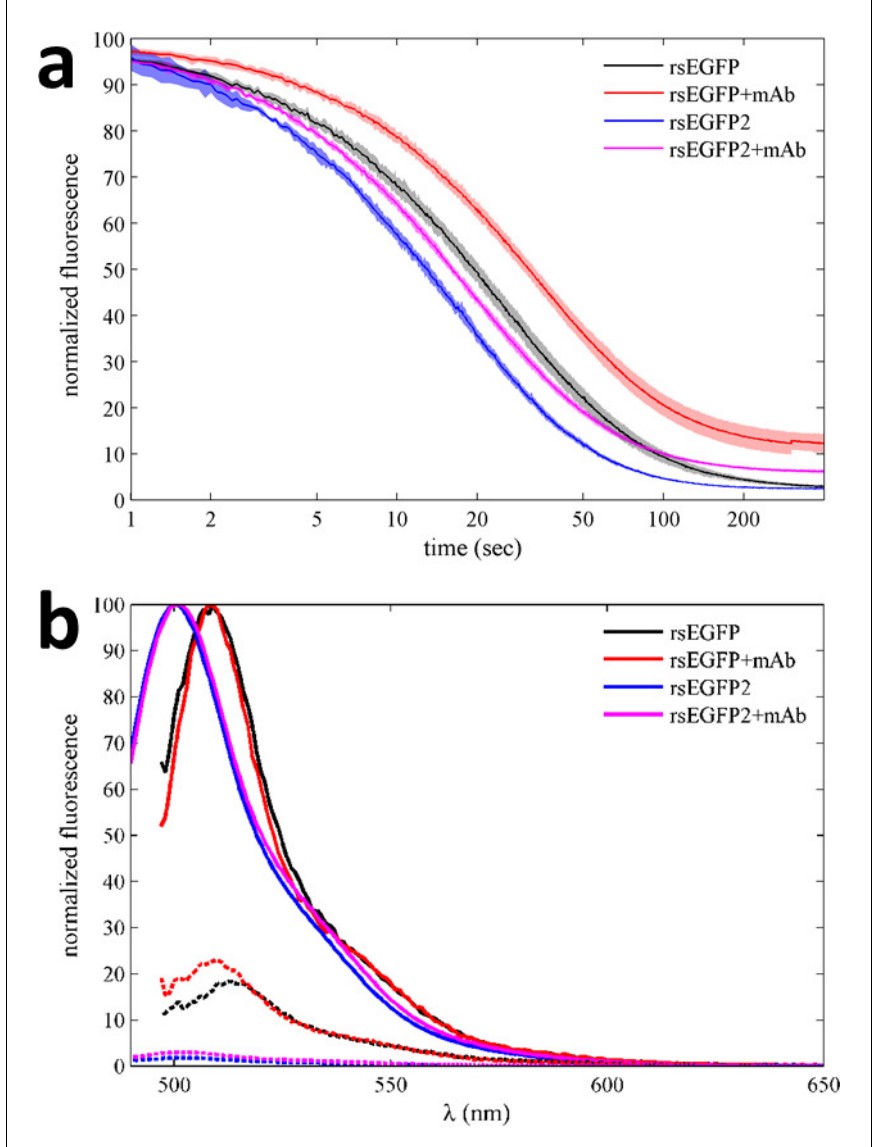

**Figure 4.** Photoswitching behavior and fluorescence emission spectra of EGFP variants in the absence and presence of GFP antibody in a benchtop spectrofluorometer. (a) Normalized fluorescence data of 500 nM rsEGFP in the presence (red) and absence (black) of equimolar anti-GFP mAb, monitored at 508 nm during excitation at 490 nm. Equivalent experiments for rsEGFP2 in the presence (magenta) and absence (blue) of the same antibody, measured at 502 nm during 483 nm excitation. The incident power density is 3 mW/cm². The data shown here represent the average of at least four measurements for each sample, with the line width (or patch width) representing the data acquisition errors. (b) Normalized fluorescence spectra of 500 nM rsEGFP with excitation at 490 nm in the presence (red) and absence (black) of equimolar anti-GFP mAb, before (solid lines) and after (dotted lines) photoswitching. Equivalent emission spectra for rsEGFP2 with 483 nm excitation in the presence (magenta) and absence (blue) of the same antibody, before (solid lines) and after (dotted lines) photoswitching.

MCMC decomposition *via* *Equation 5* results in an excellent fit (*Figure 3a*), with sedimentation coefficient distributions of species with temporal signal characteristics of fluorescein and rsEGFP2 as shown in *Figure 3c* (solid lines). The decaying component is correctly found to migrate with ~2.5 S, and the temporally stable signal is found to sediment in the usual hydrodynamically resolved monomer, dimer, and trimer peaks of BSA. The relative errors in the distinction between rsEGFP2 and fluorescein components are calculated to be 0.21% over the entire range (based on *Equation 6*).

The sedimentation coefficients extracted from the mixture are highly consistent with those obtained in control experiments of individual protein components (dashed lines in *Figure 3c*), as expected for a mixture of non-interacting species.

Fluorescence SV data from a three-protein mixture with interactions is shown in *Figure 3d*, consisting of two psFPs (rsEGFP2 and Padron), in addition to a molar excess of unlabeled monoclonal anti-GFP antibody (mAb). From visual inspection of the data one can discern a boundary structure with more complex temporal modulation, which, again, cannot be accounted for with any conventional boundary model (*Figure 3e*). Application of the MCMC decomposition produces an excellent fit, with component sedimentation coefficient distributions (*Figure 3f* solid lines) again being highly consistent with the results from the separately measured Padron and rsEGFP2/mAb mixture (*Figure 3f* dashed lines). Here, the sedimentation coefficient distribution of rsEGFP2 shows a 7 S peak with no 2.5 S peak, consistent with the formation of an antibody/rsEGFP2 complex. Thus, the binding partner of the antibody can be correctly identified by exploiting the new temporal signature dimension characteristic of the psFPs. Based on *Equation 6*, the relative error in assignment of component concentrations is 2.6% at 2.5 S and 1.1% at 7 S. Thus, these model systems demonstrate the potential for simultaneous component discrimination and hydrodynamic resolution.

## Choice of psFPs, switching kinetics and laser power

When the psFPs are involved in complex formation, the question arises whether the photoswitching kinetics of psFPs may also depend on the complex state. In fact, in modeling the data of *Figure 3d*, the time-constants of rsEGFP2 were initially constrained to the separately measured values, but subsequent refinement led to a significantly better fit: As may already be visually discerned from comparison with *Figure 3—figure supplement 1*, a portion of the fast-moving boundary of the rsEGFP2/mAb complex in *Figure 3d* remains fluorescent after long time ($\alpha$ = 0.2070 [0.2052–0.2088, 95% CI]), whereas free rsEGFP2 shows nearly complete conversion to the dark state ($\alpha$ = 0.0274 [0.0270–0.0281, 95% CI]). Clearly, when bound by the anti-GFP antibody, the contrast of rsEGFP2 is lower than in the unbound state. Similar observations were made with rsEGFP, which exhibits slightly altered photoswitching kinetics in the presence of the anti-GFP antibody (data not shown). These observations were confirmed by rsEGFP and rsEGFP2 fluorescence measurements in a benchtop fluorimeter (*Figure 4a*), where we found marked alterations in the off rates and on/off contrast ratios for both rsEGFP and rsEGFP2 in the presence of antibody as compared to their unbound state. At the same time, their emission spectral properties changed little with the addition of the antibody (*Figure 4b*). These effects point to possible conformational changes induced by antibody binding impacting the equilibrium of photophysical states (see Discussion). In general, however, such effects should be irrelevant in applications of this approach where the psFP will be designed as an inert tag rather than offering a binding site.

We have found that several psFPs are suitable for FDS-SV with well-defined photophysical characteristics under the current illumination conditions. Besides rsEGFP, rsEGFP2 and Padron these also include Dronpa (*Zhao et al., 2014b*). While older FDS instruments are equipped with a fixed-power 13 mW solid-state laser, newer models operate with an adjustable 50 mW diode laser that offers additional flexibility for modulating photoswitching kinetics by using different power settings for the scanning beam. The optimal choice of psFP and excitation power will be determined by consideration of the temporal conversion characteristics of all psFPs in the mixture, in combination with the expected time-course of sedimentation, as dictated by *s*-value of the protein complex and the rotor speed. We have implemented *Equation 6* in the data analysis software as a design tool that – prior to data acquisition – can guide the psFP selection to maximize the contrast for MCMC based on known psFP characteristics in FDS and the sedimentation coefficient range of the expected protein complexes. For example, the black line in *Figure 5* shows the theoretically best switching rate constant for a 2.8 S species under standard SV conditions to be between 1–5×10$^{-4}$/sec.

## Blinking and FRAP-like sedimentation with psFPs

It is evident that switching of psFPs is required for discrimination, but at the same time the decaying boundary amplitudes for psFPs that switch off during sedimentation will diminish the information content on such psFPs-tagged species, as the signal disappears at later times. However, rapid photoswitching can be advantageous if we exploit the reversibility of the process to switch it back on.

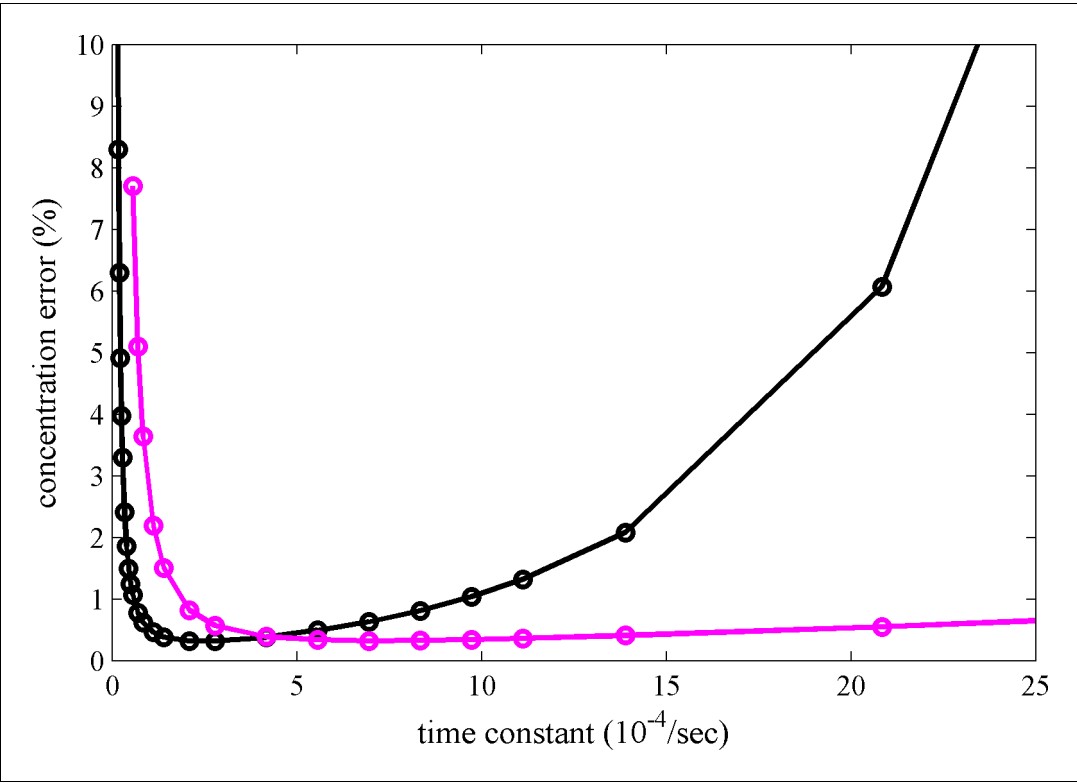

**Figure 5.** Calculated relative concentration error for distinguishing photoswitching from non-switching FPs in SV, with and without blinking. The calculations are based on a decaying signal contribution of the psFP with different rate constants, assuming standard sedimentation conditions (12 mm solution column, 20°C, 50,000 rpm, similar to data shown in *Figure 2c*) and an *s*-value of 2.8 S. Shown are the predictions of *Equation 6* for constant scanning (black) and blinking (magenta) with 2 min reset events occurring at 26 min and 69 min.

Stopping the 488 nm exposure will cause rsEGFP to spontaneously relax back into the equilibrium fluorescent state on the time-scale of an hour. Thus, passive restoration of the fluorescent signal by simply pausing the data acquisition is possible (data not shown) but presents a viable approach only for slowly sedimenting species. However, rapid and virtually complete restoration of the initial psFPs signal is possible by illumination at 405 nm. In this way, for example, rsEGFP and rsEGFP2 can undergo hundreds of on/off cycles with very little loss (*Grotjohann et al., 2012*). To this end, we took advantage of the window in the rotor chamber that usually forms part of the interference optical system, and modified it to allow illumination of the spinning rotor with a high-power 405 nm LED at ~10 mW/cm$^2$. To achieve a quantitatively uniform reset of fluorescence, a sector-shaped mask was used to produce constant 405 nm exposure times for molecules rotating through the beam at all radii. SV experiments were carried out with several cycles of rapid signal depletion under 488 nm illumination at 25 mW (producing a depletion rate of $9.7 \times 10^{-4}$/s for rsEGFP2), alternating with a 2 min pulse of 405 nm illumination (leading to > 98% recovery of the fluorescence signal of rsEGFP2). The sequence of exposure and scanning events is schematically represented in *Figure 6a*. An example for the resulting 'blinking' boundary for rsEGFP2 is shown in the *Video 4* and *Figure 2e*. An experiment with Padron exhibiting the opposite blinking behavior is shown in *Figure 6b*. With signal recovery time-points included into the model, blinking SV data could be fitted within the noise of data acquisition, and blinking has been fully implemented in the MCMC decomposition of signals from mixtures with different fluorophores. However, on the basis of the theoretical analysis of information content in *Figure 5* (magenta line), for switching processes at rates below $3 \times 10^{-4}$/sec, blinking can be detrimental to component discrimination, because it does not allow the fluorophore to ever significantly populate the dark state.

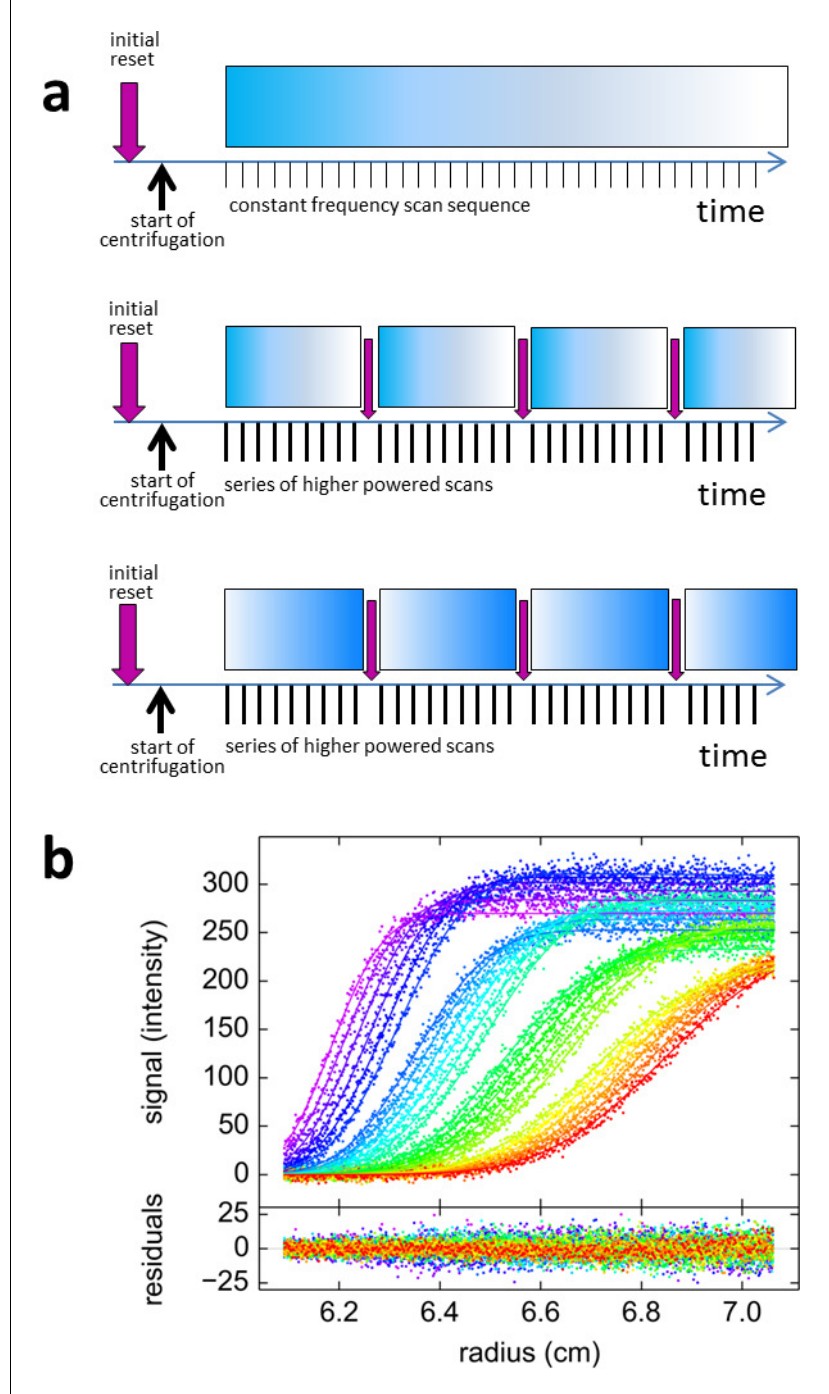

**Figure 6.** Blinking fluorescence SV data. (**a**) Cartoon illustrating the timing of events in standard *vs.* blinking experiments. Dependent on sample history, an initial brief illumination with 405 nm may be applied prior to start of centrifugation. In standard time-domain experiments (top) a series of 488 nm scans at constant frequency is initiated shortly after full rotor speed is reached. It causes a gradual depletion of signals for molecules switching off, depicted by the fading blue bar, such as shown in *Figure 2c*. For blinking experiments (middle and bottom) the sequence of scans was briefly interrupted at select time points (indicated by purple arrows) by short pulses of high-power 405 nm illumination, switching on molecules that switch off at 488 nm such as rsEGFP2 in *Figure 2e*, or switching off molecules that switch on at 488 nm such as Padron in (**b**), respectively. (**b**) Example of blinking data from 30 nM Padron sedimenting at 50,000 rpm and 20°C. Fluorescence scans were acquired with 488 nm excitation in ~5 min intervals (only every second scan shown), interrupted three times by 2 min exposure of the spinning rotor to 405 nm light. During the data acquisition cycles the fluorescence signal increases, due to

*Figure 6 continued on next page*

*Figure 6 continued*

switching on of Padron, but the increase is reversed each time in the 405 nm light that switches Padron to the off state. At these time points, gaps in the boundary overlay can be visually discerned; these appear largely due to the decreasing signal magnitude of the boundary. The complete fluorescence or dark state is not reached in these cycles, and the fluorescence signal magnitude after 405 nm exposure is treated as a fitting parameter. The solid line is the best-fit combined sedimentation/photoswitching model of a single species with apparent molar mass of 26.9 kDa and s-value of 2.58 S.

For the slow switching case, we followed a different approach to enhance discrimination by creating a spatially localized trough in the psFP fluorescence signal. In contrast to blinking, this can be executed without any instrument modification, simply by positioning the existing FDS detector at a fixed radius for an extended period of time at the beginning of the sedimentation process. Based on the size of the focal spot we estimate the stationary exposure to be on the order of 100-fold stronger as compared to the time-averaged exposure during scanning across the standard radial scan range. Localized switching to the dark state causes a trough in the fluorescence signal (or a peak for molecules switching on at 488 nm). Much like the well-known technique of fluorescence recovery after photobleaching (FRAP), molecules in the dark state that are located initially in the trough will diffuse and exchange with fluorescent molecules from outside, diminishing the trough. In contrast to FRAP, the diffusion processes are coupled to size-dependent migration from sedimentation, which translates and stretches the shape of the trough as it diminishes. Simultaneously, this process is superimposed by the same uniform photoswitching of psFPs that sets in with radial scanning (*Video 5*). Even though this appears to be a more complicated process, the evolution of the trough is naturally modeled in the same framework developed above for the sedimentation in quasi-uniform exposure: Since SV models already account for diffusional spread superimposed to sedimentation, the FRAP-like SV model solely requires the additional application of appropriate boundary conditions (*Equation 8*) at early times. However, in comparison to standard sedimentation boundaries, the resulting data are enriched in several ways, as (1) additional information on diffusion becomes available; (2) two additional sedimentation boundaries are generated to improve the precision of

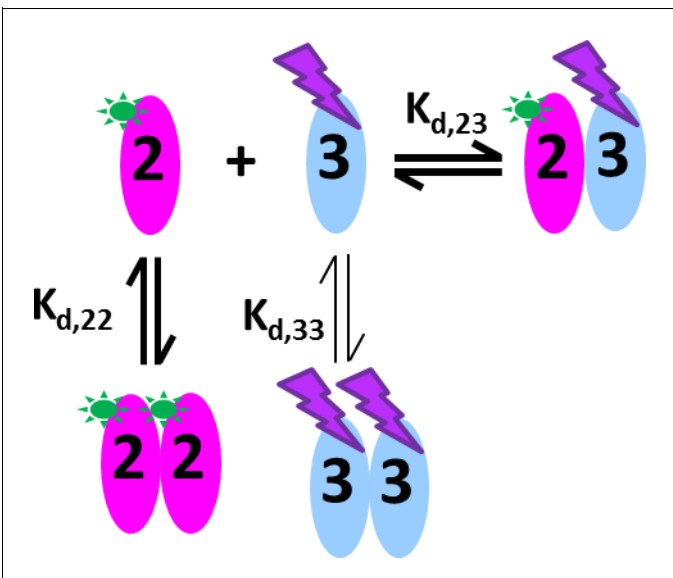

**Figure 7.** Cartoon of the mixed homo- and hetero-dimerization of GluA2-ATD and GluA3-ATD. DL488-GluA2 (magenta) exhibits moderately strong homo-dimerization, while rsEGFP2-GluA3 (blue) undergoes weak homo-dimerization. Both are competitive with a strong hetero-association. DL488-GluA2 provides a constant fluorescent signal contribution, whereas rsEGFP2-GluA3 signals are modulated and can be recognized in the time-domain of the signal.

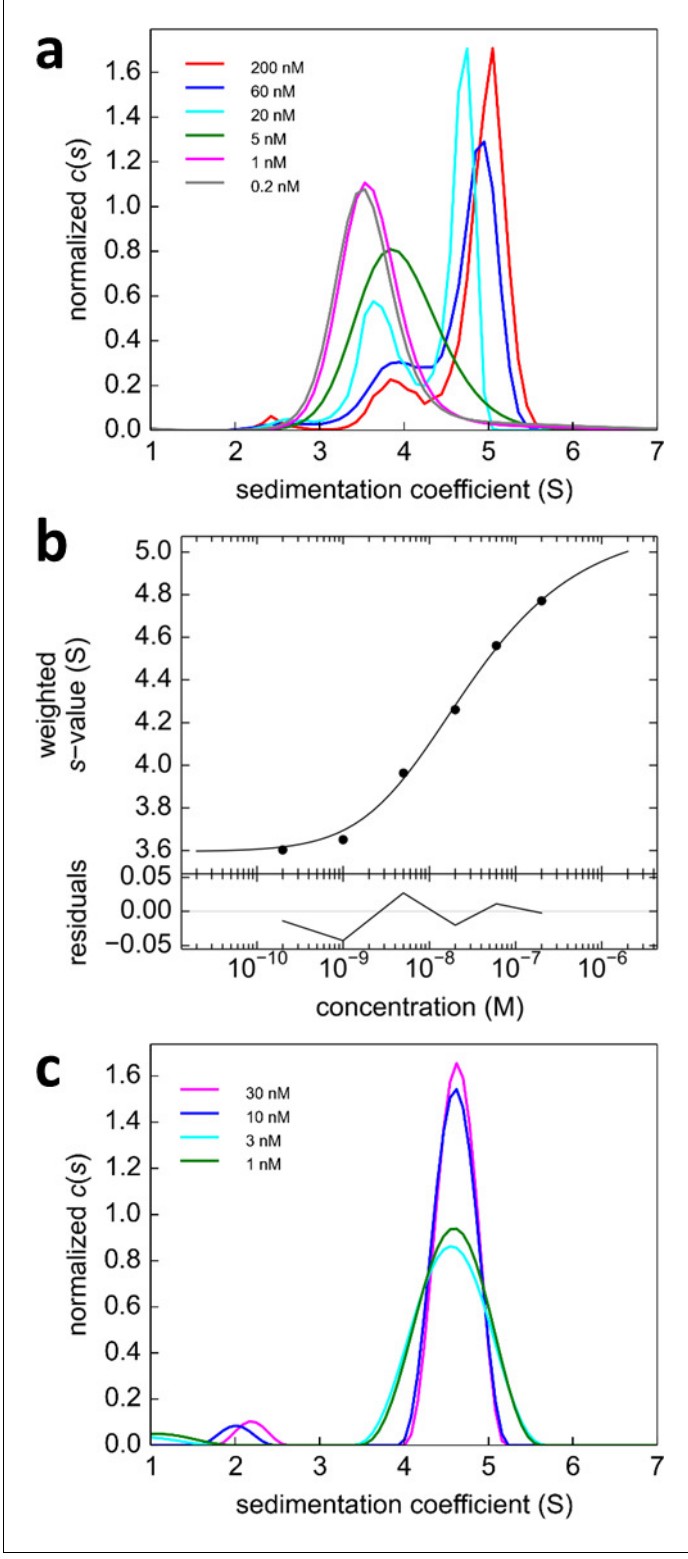

**Figure 8.** Analysis of the self-association of DL488-GluA2 and rsEGFP2-GluA3. (**a**) DL488-GluA2 samples at a series of concentrations from 0.2 nM to 200 nM were sedimented at 50,000 rpm, 20°C and FDS scans were acquired with 488 nm excitation. The resulting SV data were fit to a $c(s)$ model. (**b**) Isotherm of $s_w$-values determined by integration of c(**s**) profiles in (**a**) (circles). The monomer-dimer model of the isotherm (solid line) leads to best-fit values of $s_1$ = 3.57 S, $s_2$ = 5.11 S (uncorrected $s$-values) and a best-fit $K_{d,22}$ of 24 [9–59; 95% CI] nM. (**c**) Sedimentation coefficient distributions of rsEGFP2-GluA3 at concentrations between 1 nM and 30 nM, determined

*Figure 8 continued on next page*

*Figure 8 continued*

in blinking configuration as shown in *Figure 2e*. The constant peak in this concentration range is consistent with the previously determined $K_{d,33}$ of 5.6 µM (*Zhao et al., 2012*).

sedimentation coefficients; and (3) in data from mixtures of psFPs with stable fluorescent molecules, the depth of the trough will provide a landmark at early times reporting directly on the concentration of psFPs-tagged components. In applications to the analysis of experimental data, excellent fits were achieved with this model with residuals close to the noise of data acquisition. For example,

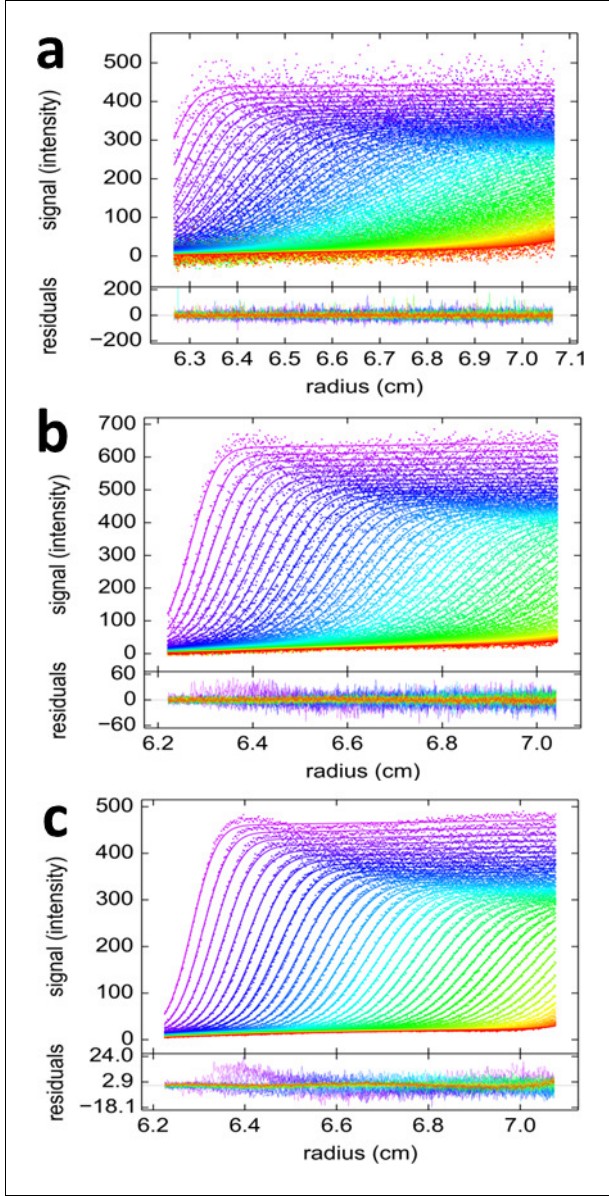

**Figure 9.** Competitive self-association and hetero-association of DL488-GluA2 and rsEGFP2-GluA3. Scan data were acquired by FDS-SV, continuously scanning with 488 nm excitation during the sedimentation at 50,000 rpm, 20°C. Data shown are for equimolar concentrations at 0.3 nM (**a**), 3 nM (**b**) and 10 nM (**c**). Solid lines and residuals are from the MCMC decomposition, with pre-determined switching parameters for rsEGFP2-GluA3, leading to distributions shown in *Figure 10a*.

with the single-species model fitted to the rsEGFP2-GluA3 data shown in *Figure 2f*, the s-value is 4.59 [4.50–4.68, 68% CI] S, consistent with the value obtained from blinking and from constant illumination experiments.

## Application to the GluA2 – GluA3 ATD interaction

Glutamate receptor ion channels (iGluRs) assemble into homo- and hetero-tetramers with their subtype composition dictating the properties of synaptic transmission (*Geiger et al., 1995*). It is thought that the strength of homo- and hetero-dimerization of the amino-terminal domains of different iGluRs governs their assembly and tetrameric architecture (*Rossmann et al., 2011*; *Kumar et al., 2011*). Therefore, the binding affinities of iGluRs have been of significant interest (*Zhao et al., 2012*; *Rossmann et al., 2011*; *Karakas et al., 2011*; *Clayton et al., 2009*; *Kumar et al., 2009*). In a pioneering study, homo- and hetero-dimerization affinities for different AMPA receptor amino terminal domains (ATDs) were determined by standard FDS-SV (*Rossmann et al., 2011*). However, for analysis of hetero-oligomerization only one component could be monitored, and the FAM-label used in that study was later found to induce artificially strong homo-dimerization of GluA2 ATD (*Zhao et al., 2013c*), with its effect on interactions with the other AMPA receptor subunit ATDs unknown. Recent work, which has resulted in structures of full length heteromeric AMPA receptors assembled from GluA2 and GluA3 (*Herguedas et al., 2016*) provides a compelling reason for accurately measuring the affinity for assembly of GluA2/GluA3 ATD heterodimers. Using Dylight488-labeled GluA2-ATD (DL488-GluA2) and a fusion protein of GluA3-ATD and rsEGFP2 (rsEGFP2-GluA3), we take advantage of the component discrimination in MCMC to monitor simultaneously the homomeric and heteromeric complexes, which consist of GluA2 monomers and homodimers, GluA3 monomers and homodimers, and GluA2/GluA3 heterodimers (*Figure 7*). In the initial control experiments, a dilution series of DL488-GluA2 (*Figure 8a,b*) yielded a homo-dimerization $K_{d,22}$

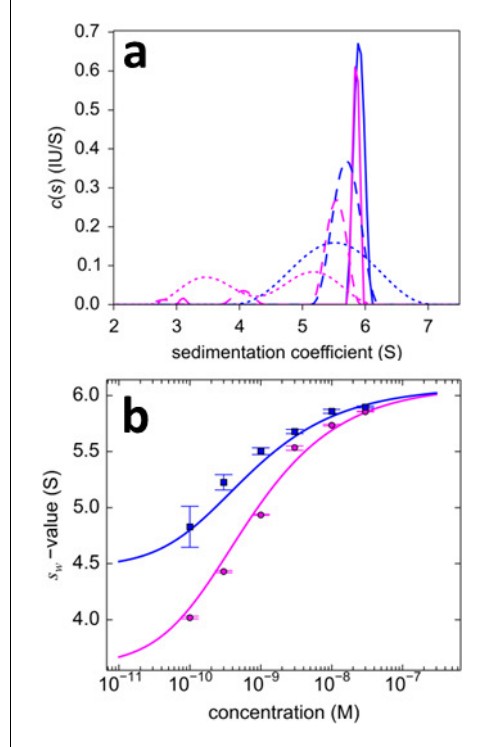

**Figure 10.** Dilution series analysis of GluA2-GluA3 interaction. Sedimentation at 50,000 rpm of a dilution series of DL488-GluA2 with equimolar rsEGFP2-GluA3 was observed with 13 mW excitation beam in FDS-SV, causing a decrease of rsEGFP2 signal with time-constant of $5.5 \times 10^{-4}$/sec. (a) Sedimentation coefficient distribution of the decaying signal component of rsEGFP2-GluA3 (blue, reporting on all blue species in *Figure 7*) and the constant signal component of DL488-GluA2 (magenta, reporting all magenta species in *Figure 7*) at 30 nM (solid lines), 3 nM (dashed lines) and 0.3 nM (dotted lines). Raw data and fits are in *Figure 9*. (b) Weighted-average sedimentation coefficients $s_w$ for the rsEGFP2-GluA3 (blue circles) and DL488-GluA2 (magenta circles), with 68% CI error bars from Monte-Carlo analysis. The solid line is the best-fit isotherm for a linked homo- hetero-dimerization equilibrium, using a fixed $K_{d,22}$ of 27.1 nM for the separately determined GluA2 homo-dimerization. The best-fit estimate for the hetero-dimerization $K_{d,23}$ is 0.32 [0.20–0.46] nM.

of 24 [9–59; 95%CI] nM; no homo-dimerization was detected for rsEGFP2-GluA3 at concentrations up to 30 nM (*Figure 8c*). These results are consistent with previously measured $K_{d22}$ and $K_{d33}$-values for the homo-dimerization of GluA2 and GluA3, 20.5 nM [15.9–26.4; 95% CI] (*Zhao et al., 2013a*) and 5.6 [1.7–14] µM (*Zhao et al., 2012*), respectively, while by contrast, the homo-dimerization $K_{d,22}$ of 1.8 nM for FAM-labeled GluA2 was 11-fold lower (*Rossmann et al., 2011*).

To analyze the heterogeneous interaction between GluA2 and GluA3 ATDs, an equimolar dilution series from 0.1–30 nM was studied in a configuration with a 13 mW excitation beam, distinguishing rsEGFP2-GluA3 from DL488-GluA2 solely from slow photoswitching of the former. Representative raw

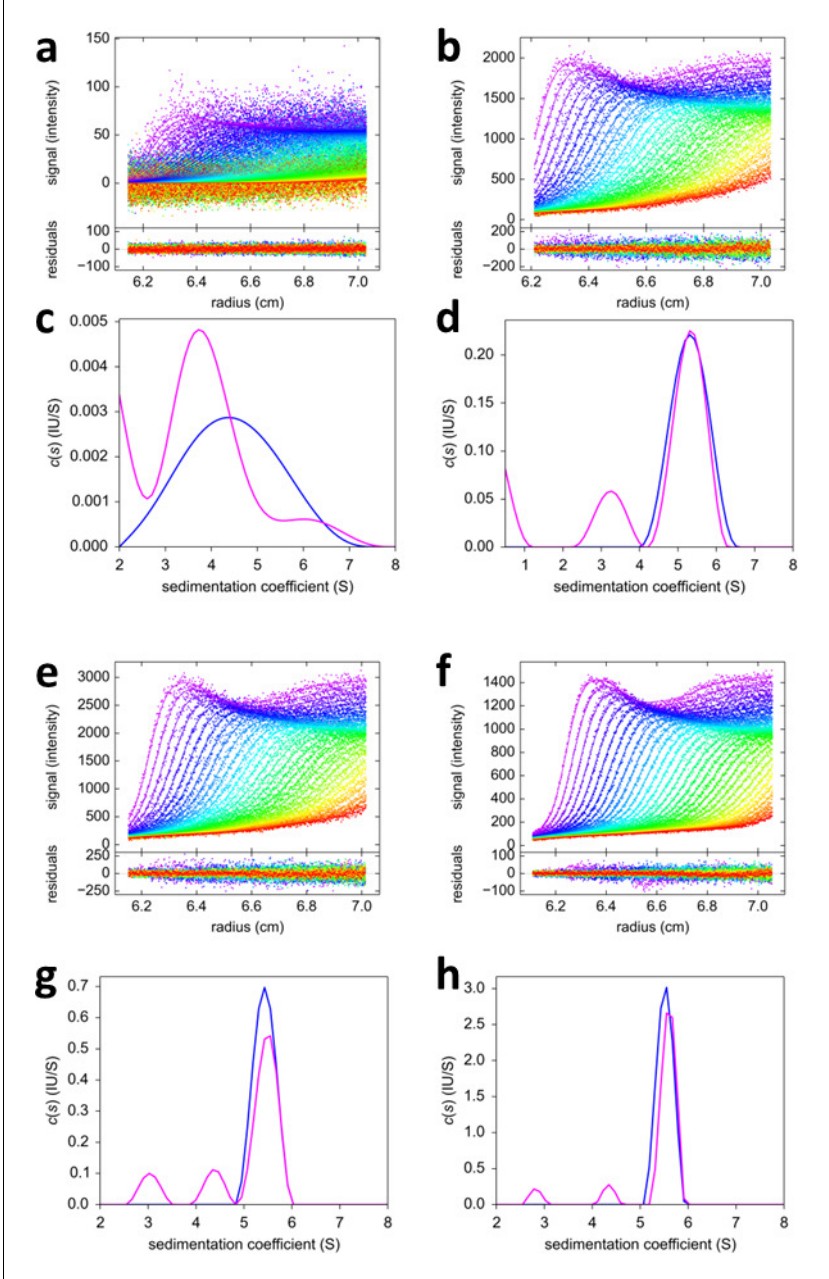

**Figure 11.** FRAP-like FDS-SV from mixtures of rsEGFP2-GluA3 and DL488-GluA2. Equimolar mixtures at 0.1 nM (**a, c**), 2 nM (**b,d**), 3nM (**e,g**) and 10 nM (**f,h**) were sedimented at 50,000 rpm, 20°C, with initial 20 min stationary exposure at 6.5 cm of the excitation beam at 488 nm, prior to radial scanning with 488 nm excitation (every second scan shown). Initial exposure parameters were taken from the rsEGFP2-GluA3 data in *Figure 2f* and fixed. Solid lines and residuals are from the MCMC decomposition (**a,b,e,f**) with resulting distributions shown below the scans in panels (**c,d,g,h**), with the rsEGFP2-GluA3 component in blue and the DL488-GluA2 component in magenta. DOI: 10.7554/eLife.17812.019

data sets and MCMC fits to the sedimentation boundaries are shown in *Figure 9*, leading to calculated component sedimentation coefficient distributions shown in *Figure 10a*. At a high concentration of 30 nM, the sedimentation coefficient distributions rsEGFP2-GluA3 and DL488-GluA2 both show a single co-sedimenting peak at ~5.9 S, consistent with a saturated 1:1 hetero-dimer complex, moderately extended with a frictional ratio of 1.42. With decreasing concentrations the peaks occur at lower sedimentation coefficients, characteristic for a rapid association/dissociation equilibrium relative to the

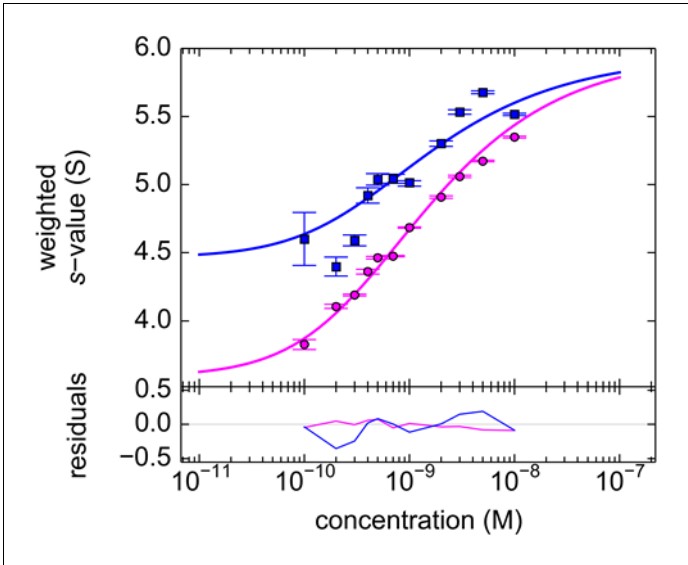

**Figure 12.** Component $s_w$ isotherm of DL488-GluA2 and rsEGFP2-GluA3 as observed in FRAP-like FDS-SV. Equimolar mixtures were studied by FDS-SV with initial localized switching, producing data including those shown in *Figure 11*. Plotted here are component $s_w$-values for rsEGFP2-GluA3 (blue) and DL488-GluA2 (magenta) from integration of the MCMC component sedimentation coefficient distributions. Error bars shown are estimated 68% confidence intervals from Monte-Carlo analysis. The model of competitive homo- and hetero-dimerization (lines) results in an estimate of the heteromer $K_{d,23}$ of 0.68 [0.36–1.24, 95% CI] nM.

time-scale of sedimentation (*Schuck, 2010*). Due to the decreasing signal/noise ratio peaks also broaden at lower concentrations. Further, at concentrations below 1 nM a separate slower sediment-ing peak of DL488-GluA2 can be discerned, consistent with the s-value of free GluA2 monomer, co-existing with a faster-sedimenting dimeric fraction of GluA2. Due to the smaller difference between the s-value of the hetero-dimeric complex and free rsEGFP2-GluA3, the latter cannot be resolved at the low signal/noise ratios below 1 nM. However, the signal-weighted average sedimentation coeffi-cient $s_w$ can still be determined as a function of concentration for both components (*Figure 10b*). The Monte-Carlo error analysis shows $s_w$ to be well-defined at all concentrations for the constant signal component, but for the decaying signal component statistical errors increase at lower concentrations, with the lowest useful data point at 0.3 nM. A global fit with an isotherm model for the coupled system where hetero-oligomerization of GluA3 is competitive with homo-oligomerization of GluA2 leads to an estimate of the heterodimer $K_{d,23}$ of 0.32 [0.20–0.46, 95% CI] nM, with a limiting s-value for the complex of 6.09 (6.01–6.17) S.

To explore an alternative MCMC design, an experiment with the FRAP-like SV configuration was carried out (*Figure 11*). Mixtures of rsEGFP2-GluA3 and DL488-GluA2 were exposed to localized 488 nm light from the stationary FDS for a period of 20 min. The initial illumination was assessed from the analysis of a control psFP sample spinning side-by-side in the same rotor sharing the same exposure (*Figure 2f*), and then fixed for the analysis of the mixtures (*Figure 11*). Visually, the reduced trough depth directly reflects the smaller relative signal contribution of rsEGFP2. Consistent with the results from the previous experimental series, the weight average component sedimentation coefficients series from the FRAP-like SV leads to an estimate of the heteromer $K_{d,23}$ of 0.68 [0.36–1.24, 95% CI] nM (*Figure 12*).

## Discussion

The present work shows the proof of principle for how different macromolecular components can be readily distinguished in fluorescence-based sedimentation velocity experiments using single wave-length excitation in a way that is naturally compatible with many fluorescent probes commonly used in super-resolution microscopy. Although psFPs were designed to serve a very different purpose, we

have embarked on the quantitative analysis of their switching kinetics as a novel temporal tag to substitute for spectral discrimination. In the low-power illumination conditions of FDS-SV the time-course of photoswitching can serve as a finger-print for different psFPs and permit quantitative multi-component analysis. Thus, the study of heterogeneous interactions can be carried out with mixtures of psFP-tagged proteins, proteins labeled with stable fluorophore dyes, and/or non-fluorescent proteins. This opens the possibility for the same fusion proteins observed co-localizing in microscopy to be overexpressed and purified for in vitro binding studies in FDS-SV, to provide information of number, size, and stoichiometry of complexes. Further, there is a potential for FDS-SV to be applied to more crowded solutions such as cell extracts and serum (*Polling et al., 2013*; *Kokona et al., 2015*; *Hill and Laue, 2015*), in which the hydrodynamic interpretation of complex sizes is less quantitative, but complex stoichiometries derived from multi-component detection should be invariant. Exploiting the switchable signal further, we have demonstrated that additional spatial and temporal modulation of the psFPs signal in FDS-SV is possible. This has the potential to further improve discrimination of different psFPs and to enhance hydrodynamic analysis, for example, in the FRAP-like configurations used in the present study.

The switchability of signal and localized optical manipulation afforded by psFPs can be envisioned to expand the toolbox of analytical ultracentrifugation in many other unexpected ways. For instance, the creation of lamella-like signal configurations within the concentration plateau region may enable novel assays, for example, to observe subunit exchange rates of size-resolved macromolecular complexes from the lamella migration and signal amplitude changes, or to study complex formation in crowded solutions within the plateau region of rapidly sedimenting co-solutes. Even though the present work was focused on SV, sedimentation equilibrium (SE) analytical ultracentrifugation would be expected to equally benefit from the use the new temporal tag: for example, once molecules are in thermodynamic equilibrium of sedimentation, psFPs may be reset or depleted, for example, by uniform 405 nm or 488 nm illumination, and the ensuing temporal signal evolution could reveal the equilibrium distribution specifically of psFPs in a mixture. Further, one could envision FRAP-like localized switching in SE to complement thermodynamic with hydrodynamic information obtained while maintaining SE. Beyond analytical ultracentrifugation, quantitatively exploiting the characteristic dynamics of photoswitching arising from different switching quantum yields may prove useful in other techniques.

The new MCMC approach we developed may be compared to other fluorescence methods used to detect binding, including fluorescence correlation/cross-correlation, polarization, proximity and life time imaging methods. These generally offer greater flexibility in solution conditions and can be carried out in vivo. However, the strongly size-dependent migration of molecules in the gravitational field is a unique feature of FDS-SV that can lead to significantly higher resolution of macromolecular complexes. We believe this has great potential for the study of systems that exhibit polydisperse mixtures of different complexes, which would elude traditional structural characterization, and only display population-averaged features in spectroscopic techniques.

Of interest, we discovered changes in the switching rate and final contrast of rsEGFP and rsEGFP2 induced by antibody binding. This suggests that subtle binding-induced conformational changes of the GFP β-barrel occur that can lead to changes in the equilibrium populations of photophysical states of the chromophore. For Dronpa, NMR studies have revealed flexibility of the β-barrel allowing non-radiative decay processes in the dark state (*Mizuno et al., 2008*), and viscosity-mediated reduction of flexibility has been proposed as a mechanism for the viscosity sensitivity of its fluorescence (*Kao et al., 2012*). Also, structural flexibility mediating conformational changes from the protein exterior to the chromophore environment inside the β-barrel has been exploited for GFP as a sensor (*Berg et al., 2009*). It seems conceivable that a similar mechanism may be at work in the binding sensitivity of its photoswitchable variants. Even though the psFPs will typically be designed to be an inert tag, rather than offering binding sites, this reveals the potential for an additional, more subtle source of information in temporally modulated FDS-SV, if the equilibrium of photophysical states is altered through occupation of a binding site.

Prior work on the assembly of AMPA receptor ATDs suggested an inefficient process for the formation of GluA2/GluA3 heteromers because the $K_{d,22}$ for homodimer assembly by GluA2, 1.8 nM, was only 1.4 fold greater than the $K_{d,23}$ of 1.3 nM for heterodimer assembly (*Rossmann et al., 2011*). Using the MCMC approach we find that the process is much more efficient, with a $K_{d,23}$ for heterodimer assembly of 0.32 nM, 85-fold smaller than the 27.1 nM $K_{d,22}$ for homodimer assembly

by GluA2, and four orders of magnitude smaller than the $K_{d,33}$ for GluA3 homo-dimerization, which is consistent with the obligate formation of GluA2/GluA3 heterodimer in neurons (*Rossmann et al., 2011*).

While this first application highlights the potential of the method to determine sub-nanomolar dissociation equilibrium constants of protein complexes involving competitive or cooperative self- and hetero-association processes, other key aspects of the method are the ability to simultaneously monitor the size-distribution of individual components and their complexes formed in solution, and the potential to discriminate two or more different psFPs. The availability of size-distributions for all components distinguishes MCMC FDS-SV from methods such as SPR or ITC often used for studying bimolecular reactions. Considering that, from large-scale proteomic experiments (*Gavin et al., 2002*; *Krogan et al., 2006*), on average more than four components are in cellular protein complexes, it is necessary to develop methods that can elucidate their stoichiometries, architectures, and driving forces. We believe MCMC FDS-SV offers a powerful new approach to study such multi-component complexes in solution.

## Materials and methods

### Analytical ultracentrifugation with fluorescence detection system (FDS)

SV experiments were carried out in an Optima XL-A (Beckman Coulter, Indianapolis IN) equipped with an FDS (Aviv Biomedical, Lakewood NJ). The FDS light source is a 488 nm 10 mW diode laser, and emission is detected between 505 nm and 565 nm. Setup of the AUC followed standard protocols (*Zhao et al., 2013b*), using an 8-hole An-50 TI rotor. In order to minimize local exposure of the samples during FDS adjustment and magnet angle locking, radial calibration and PMT settings were evaluated at 3000 rpm, followed by temperature equilibration at 20°C, and acceleration to 50,000 rpm. If initial FDS adjustment required more than a few minutes, the samples in the rotor were illuminated with 405 nm prior to temperature equilibration. With the FDS focal depth set at 4000 μm and the PMT power at 51%–80% with gain of 8 (dependent on sample concentration), sequences of radial scans were acquired continuously for 12 hr with radial intervals of 20 μm.

For passive blinking experiments, pauses in data acquisition were applied. For rapid blinking experiments a modified Optima XL-A was used, where 405 nm light from a high-power LED was guided into the rotor chamber through the interference optics port in the heat sink to illuminate the spinning rotor. A sector-shaped mask concentric to the axis of rotation was used.

### Photoswitching and exposure geometry in FDS

The FDS signal is linear with concentration in the nanomolar concentration range and below (*Zhao et al., 2013b*; *Lyons et al., 2013*), and accordingly, we consider the fluorescence signal $F(r,t)$ a product of concentration $c(r,t)$ and a specific molar fluorescence signal increment $E$. The overall initial signal increment $\epsilon_0$ will depend, for example, on laser power, absorption coefficient, intrinsic fluorescence quantum yield, and detection efficiency. Photoswitching is modeled as a two-state process, with a single exponential transition from an initial equilibrium of photophysical states to another equilibrium after a long time of 488 nm irradiation in the FDS. The rate of transition will depend on a quantum efficiency of photoswitching and the total count of photons incident on the migrating fluorophore. The latter will be dependent on the incident laser power density, as well as the distance from the center of rotation.

Briefly, since all molecules have the same angular velocity $\omega$, they traverse a constant beam width $\delta$ more quickly at higher radii. At position $r'$ the sample has a transverse velocity $\omega r'$ and (with negligible difference between the arc and secant in the AUC geometry) molecules are illuminated for the time $\delta/\omega r'$ at each rotation. It follows that their total photon count is

$$\Phi(r,t) = \phi \int_0^t \frac{\delta}{2\pi r'(t')} dt' \tag{1}$$

where $r'(t)$ is the trajectory of the molecule through the sample. Neglecting Brownian motion, the path history of particles with sedimentation coefficient $s$ found at time $t$ at position $r$ is

$$r'(t') = re^{\omega^2 s(t'-t)} \tag{2}$$

which, after integration, leads to

$$\Phi(r,t) = \phi \frac{\delta}{2\pi r \omega^2 s}\left[e^{\omega^2 st} - 1\right] \tag{3}$$

as the accumulated total photon count seen by the fluorophore found at radius $r$ at time $t$ sedimenting with $s$. Therefore, combining constants from illumination and molecular photoswitching into a single parameter $\beta$, we find the time-dependent fluorescent signal increment

$$E(s,r,t-t_{405,i}) = \varepsilon_0\left(1 + \alpha\, exp\left\{-\frac{\beta}{(r/r_0)\omega^2 s}\left[e^{\omega^2 st\left(t-t_{405,i}\right)} - 1\right]\right\}\right) \tag{4}$$

where $\alpha$ describes the fractional signal change after long time, and $r_0$ is a reference radius. The reference time-point $t_{405,i}$ is the last time of fluorophore reset with strong 405 nm exposure (and $t_{405,I} = 0$ without reset illumination). The slight radial dependence causes slopes that for individual psFPs are indistinguishable from the effects of a small mismatch between the focal plane of the scanner and the plane of rotation causing small gradients of signal magnification (*Zhao et al., 2013b*; *Zhao et al., 2014b*,) but for studying mixtures of psFPs they need to be quantitatively accounted for.

### Monochromatic multi-component analysis

Analogous to the multi-wavelength sedimentation coefficient distribution analyses (MSSV) (*Balbo et al., 2005*), we model the evolution of experimental fluorescence scans $a(r,t)$ as a superposition of signals from different components $p$, each exhibiting a different unknown distribution of sedimentation coefficients $c_p(s)$

$$a(r,t) = p\sum\int c_p(s)E^{(p)}(s,r,t)\chi_1(s,r,t)ds \tag{5}$$

where $E^{(p)}(s,r,t)$ denotes the temporal signal change (*Equation 4*), and $\chi_1(s,r,t)$ denotes the evolution of concentration for a species with sedimentation coefficient $s$ in the centrifugal field, as calculated by the master partial differential equation for sedimentation and diffusion in the centrifugal field, the Lamm equation (*Lamm, 1929*; *Brown and Schuck, 2008*). Sedimentation and diffusion are both governed by the same hydrodynamic frictional coefficient (*Cheng and Schachman, 1955*), which allows diffusion to be estimated for all species based on a hydrodynamic scaling law $D(s)$ for particles with given translational friction coefficient (*Schuck, 2000*). Since $f/f_0$ measures the hydrodynamic extension of a particle relative to a compact sphere of the same mass and takes a narrow range of values for most folded proteins (*Serdyuk et al., 2007*; *Cantor and Schimmel, 1980*), it was fixed in MCMC. The distributions $c_p(s)$ are calculated after discretization into a linear least squares problem with Tikhonov regularization (*Balbo et al., 2005*; *Schuck, 2000*), and implemented in the public domain software SEDPHAT (https://sedfitsedphat.nibib.nih.gov/software/default.aspx). Analogously to MSSV, the approach is compatible with mass conservation-based regularization approaches to further enhance component resolution (*Brautigam et al., 2013*). After determination of $c_p(s)$, integration can be carried out to determine signal weighted-average sedimentation coefficients $s_{p,w}$ compatible with the transport method, and further isotherm analysis of $s_{p,w}$ as a function of protein concentrations can take place with binding models (*Schuck et al., 2003*, *2015*). For the GluA2/GluA3 interaction analysis, the isotherm modeling was based on the competitive self- and hetero-dimerization model in SEDPHAT.

### Error analysis

As an experimental design tool an analytical prediction of errors in the MCMC decomposition was implemented for two species with fluorophores $p$ and $q$ sedimenting at the same $s$-value, in the absence of initial localized switching. After discretization of *Equation 5* and neglect of weak $s$-value dependence of the signals, it can be shown that, given $n$ meaningful data points with relative error $\delta_a$, the relative error in $c_p$ is

$$\delta_c \approx cond(\mathbf{E}) \frac{2\delta_a}{\sqrt{n}} \qquad (6)$$

, i.e., is dependent on the condition number of the matrix $\mathbf{E}$ where

$$E_{p,q} = \sum_t E^{(p)}(t) E^{(q)}(t) w(s,t) \qquad (7)$$

contains the mutual products of signals at experimentally available scan times $t$, weighted with a sedimentation term $w(s,t) = \sum_r \chi_{s,r,t}^2$. Mathematically, $\mathbf{E}$ plays the same role as the extinction coefficient matrix in multi-wavelength analysis. Essentially, fluorophores will be distinguishable on the basis of their temporal signature if cross-terms in $\mathbf{E}$ are small during observed times in the sedimentation experiment.

For the error analysis of experimental data, a Monte-Carlo statistical analysis was implemented in order to determine error estimates for the distributions $c_p(s)$ and their integrals such as $s_w$. In the present work, 500 iterations were chosen to estimate the sensitivity of $s_w$ to data acquisition noise, not including variation of the meniscus.

## Sedimentation model for FRAP-like SV with initial localized illumination

In the presence of inhomogeneous illumination, the spatially uniform time-dependent extinction coefficient model of *Equation 4* must be extended to account explicitly for sedimentation and diffusion of molecules in fluorescent state $\chi_f$ and dark state $\chi_d$. In the presence of illumination with a spatial-temporal intensity profile $I(r,t)$ at a wavelength of 488 nm, transition from the fluorescent to the dark state occurs with a rate $k_{f-d}(r,t) = q_{f-d}I(r,t)$ (assuming a quantum efficiency $q_{f-d}$). The evolution of concentrations of molecules in the fluorescent and dark state is then described with the coupled Lamm equation (which may be extended similarly for the reverse transition)

$$\frac{\partial \chi_f}{\partial t} = -\frac{1}{r}\frac{\partial}{\partial r}\left(\chi_f s \omega^2 r^2 - D\frac{\partial \chi_f}{\partial r} r\right) - \chi_f k_{f-d}(r,t)$$
$$\frac{\partial \chi_d}{\partial t} = -\frac{1}{r}\frac{\partial}{\partial r}\left(\chi_d s \omega^2 r^2 - D\frac{\partial \chi_d}{\partial r} r\right) + \chi_d k_{f-d}(r,t) \qquad (8)$$

The spatial illumination profile $I(r,t)$ was empirically modeled as a superposition of beams with Gaussian intensity profile, accounting for the cone-shape intensity profile. The amplitudes and center radii were introduced as additional parameters refined from the fit. After initial localized illumination, spatially uniform scanning governs the further evolution of the signal of the fluorescent species and $E(s,r,t) \times \chi_f(r,t)$ can be inserted in the multi-component decomposition of *Equation 5*.

Modeling the coupled sedimentation/diffusion/photoswitching process will account for sedimentation and diffusion that takes place during strong illumination, which causes the precise shape of the trough to slightly depend on the rates of macromolecular sedimentation and diffusion. However, the initial exposure is precisely identical for all different samples in the rotor. With this approach, the precise initial illumination profile is inconsequential because the parameters of interest are only in the further evolution with time.

## Protein expression and preparation

The plasmid pQE31-rsEGFP was kindly provided by Stefan Jakobs (Max Planck Institute for Biophysical Chemistry). The plasmid rsEGFP2 was produced by mutating the pQE31-rsEGFP plasmids using following primers, T65A (5'-CCACCCTGGGCCTACGGCGTG-3'), A150V (5'-GCCACAACGTCTATATCATGG-3'), and N205S (5'-GCACCCAGTCCAAGCTGAGC-3'). Mutations are underlined. The plasmid pQE31-Padron was also provided by Stefan Jakobs and EGFP was expressed from the pRSETA-EGFP plasmid both described previously (*Zhao et al., 2014b*).

The GluA2 ATD cDNA, residues Met1–Ser404, was cloned into the pRK5-IRES-EGFP mammalian expression vector with a C-terminal LELVPRGS-His$_8$ linker, thrombin cleavage site and affinity tag. The rsEGFP–GluA3 ATD fusion construct was created as follows: The cDNA for the GluA3 signal peptide residues Met1–Gly22, followed by an SGSG tetrapeptide linker, was inserted upstream of the cDNA for rsEGFP residues Val2–Lys239, and connected by an SGS tripeptide linker to the cDNA for GluA3, residues Gly23–Ser408, followed by a C-terminal LELVPRGS-His$_8$ linker, thrombin cleavage site and affinity tag, cloned into the pRK5 mammalian expression vector. Both constructs were

expressed in HEK293T cells by transient transfection of suspension cultures using PEI MAX 40000 (Polysciences Inc.). The secreted constructs with native complex glycosylation were concentrated by ultrafiltration using Pellicon 10 kDa MW cutoff cassettes (Millipore) and purified by immobilized metal affinity chromatography using a HiTrap NTA column (GE Healthcare), followed by proteolysis with thrombin, and then ion exchange chromatography using a HiTrap Q column (GE Healthcare). The pooled fractions were concentrated and then dialyzed against AUC or labeling buffer as appropriate (150 mM NaCl, 1 mM EDTA, 20 mM NaPhosphate, pH 7.5 and 7.0 respectively).

A 14.4 μM concentration of purified GluA2 ATD was mixed with a 12 M excess of $N$-hydroxysuccinimide (NHS) ester–activated Dylight488 (Thermo Fisher Scientific) dissolved in DMSO and then diluted with labeling buffer. The reaction was incubated in the dark at room temperature for 45 min and then loaded onto a high resolution size exclusion chromatography column (Superdex 75 10/300 GL) equilibrated with AUC buffer to separate free dye from labeled protein. The protein concentration and labeling ratio were then determined by UV-Vis spectrophotometry using values for $\varepsilon_{280}$ of 55,720 $M^{-1}cm^{-1}$ for the unmodified protein, $\varepsilon_{280}$ of 10,290 $M^{-1}cm^{-1}$ and $\varepsilon_{493}$ of 70,000 $M^{-1}cm^{-1}$ for Dylight488.

Bovine serum albumin (BSA) was acquired from Sigma Aldrich (catalog#A7030, St. Louis) and fluorescently labeled with FITC using the labeling kit from Thermo Fisher targeting the primary amine on BSA (catalog#53027). Monoclonal anti-GFP IgG (D153-3) was purchased from MBL International (Woburn, MA) and subjected to exhaustive dialysis with the working buffer, phosphate buffered saline (5.62 mM $Na_2HPO_4$, 1.06 mM $KH_2PO_4$, 154 mM NaCl, pH 7.40). To all samples, unlabeled BSA was added at 0.1 mg/ml to suppress surface adsorption of the proteins of interest.

### Fluorescence spectroscopy

Benchtop fluorescence spectroscopy experiments were carried out in a Fluorolog spectrofluorometer (Horiba Instruments Inc., Irvine CA). rsEGFP or rsEGFP2 at were studied at a final concentration of 0.5 μM in the presence or absence of equimolar anti-GFP antibody, after 2 hr incubation of the mixture. Emission spectra were collected with slit widths set to 1 nm. For kinetic experiments of photoswitching, the 490 nm excited 508 nm fluorescence of rsEGFP or the 483 nm excited 502 nm fluorescence of rsEGFP2 was monitored, respectively.

## Acknowledgements

We thank Stefan Jakobs and Stefan Hell (Max Planck Institute for Biophysical Chemistry) for providing pQE31-rsEGFP plasmid. This work was supported by the Intramural Research Programs of the National Institute of Biomedical Imaging and Bioengineering, and the National Institute of Child Health and Human Development, at the National Institutes of Health, United States.

## Additional information

### Funding

| Funder | Grant reference number | Author |
| --- | --- | --- |
| National Institute of Biomedical Imaging and Bioengineering | ZIA EB000051-09 LCIM | Peter Schuck |

The funders had no role in study design, data collection and interpretation, or the decision to submit the work for publication.

### Author contributions

HZ, Conceived of the method, Designed and carried out the AUC experiments, Implemented computational analysis, Drafting or revising the article; YF, Designed and carried out the benchtop fluorescence experiments, Analysis and interpretation of data; CG, Designed fluorescent proteins and carried out protein expression and purification, Contributed unpublished essential data or reagents; EJAA, Carried out protein expression and purification, Contributed unpublished essential data or reagents; MLM, Designed fluorescent proteins and carried out protein expression and purification, Drafting or revising the article, Contributed unpublished essential data or reagents; GP,

Designed fluorescent proteins and carried out protein expression and purification, Designed and carried out the benchtop fluorescence experiments, Drafting or revising the article, Contributed unpublished essential data or reagents; PS, Conceived of the method, Designed and carried out the AUC experiments, Implemented computational analysis, Wrote the manuscript

**Author ORCIDs**

Peter Schuck, http://orcid.org/0000-0002-8859-6966

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
