## [Decision Letter]

Thank you for submitting your article "Monochromatic Multicomponent Fluorescence Sedimentation Velocity for the Study of High-Affinity Protein Interactions" for consideration by *eLife*. Your article has been reviewed by three peer reviewers, one of whom is a member of our Board of Reviewing Editors and the evaluation has been overseen by John Kuriyan as the Senior Editor.

The reviewers have discussed the reviews with one another and the Reviewing Editor has drafted this decision to help you prepare a revised submission.

Summary:

This work addresses an important experimental challenge: the quantitative characterization of the energetics and dynamics of macromolecular interactions in complex multicomponent systems. The authors describe a new method to extract multi-component information from fluorescence-based sedimentation velocity data based on definable fluorescence bleaching or photoswitching rates of fluorophores. They present an analytical framework that enables a quantitative analysis of the data and show that their method can be used to measure equilibrium dissociation constants of homo- and heterodimeric GluA2/GluA3 complexes. The manuscript describes a very clever and creative advance and further extends from the large leaps of innovation in this methodology driven by Peter Schuck over the last 15-20 years.

Essential revisions:

The data and their analysis are of high quality and rigor. The method represents quite a step forward and may be applicable to a wide range of problems. The manuscript, however, should be improved to increase its accessibility to a broad audience. At many instances, terminology is used that is likely not familiar to most readers and specialized knowledge and is implicitly assumed to be present. The authors should careful go through their manuscript and add/revise liberally to make sure it becomes accessible to a wide audience without sacrificing detail and rigor.

- In Figure 2, polydispersity is determined using the new method. Perhaps the power of this result could be more clearly illustrated if the data is presented side-by-side with a "traditional" analysis – such as c(s) fitted with radial independent noise to account for bleaching. Also the labels spec 1 and spec 2 could be labeled more clearly as to what they mean – this is not immediately apparent.

- The manuscript could be more easy to follow with some additional simple diagrams of the concepts or methodological design added to the figures. For example:a) small cartoon of the binding model and how the two signal components relate to this binding model in Figure 8) Figure 5: Were scans taken at regular intervals? If so would be useful to show on graph when the 2 min exposure happened. A diagram of the timing of events would be useful.c) Figure 1: hard to see different scans – perhaps show less scans? Also what are the purple scans? Are these run in parallel AUC cells. Also it is not clear what the sequence of events are in collecting this data. Perhaps a diagram showing the temporal series of events could be used to illustrate the concept more clearly next to the data.

- Subsection “Application to GluA2 – GluA3 ATD heteromers”: the GluA3 data should be shown for completeness, perhaps added onto Figure 6.

---

## [Author Response]

*Essential revisions:*

*The data and their analysis are of high quality and rigor. The method represents quite a step forward and may be applicable to a wide range of problems. The manuscript, however, should be improved to increase its accessibility to a broad audience. At many instances, terminology is used that is likely not familiar to most readers and specialized knowledge and is implicitly assumed to be present. The authors should careful go through their manuscript and add/revise liberally to make sure it becomes accessible to a wide audience without sacrificing detail and rigor.*

We greatly appreciate the positive feedback on the quality and utility of the method, and agree that it would greatly benefit from better accessibility to a wider audience. We made significant efforts to achieve this goal.

We have introduced a new Figure 1 showing the basics of a sedimentation experiment, in order to clarify some of the indispensable operational terms, such as ‘boundary’, ‘plateau’, and ‘sedimentation coefficient’. It has the additional advantage of seamlessly highlighting the unique potential of this technique for studying protein interactions. This distils the essence of the technique and as such is a good starting point for experts and non-experts alike. The introduction was slightly expanded accordingly.

Further, we have modified the manuscript to eliminate references to more technical aspects (such as to the Lamm equation, frictional ratios, and hydrodynamic scaling parameters) that are not central to the new method. We found these aspects can be covered well in the methods section instead, alongside the other technical information, with some additional explanatory information and references.

- In Figure 2, polydispersity is determined using the new method. Perhaps the power of this result could be more clearly illustrated if the data is presented side-by-side with a "traditional" analysis – such as c(s) fitted with radial independent noise to account for bleaching. Also the labels spec 1 and spec 2 could be labeled more clearly as to what they mean – this is not immediately apparent.

We agree this is a very useful addition for highlighting the characteristics of the psFP signals in a way we had previously not thought of.

As suggested, we have now included new panels to this Figure (now Figure 3) to show how a traditional analysis fails, even if one were to try capturing the decreasing plateau signals with time-dependent baseline offsets. (Although radial-invariant time-dependent baseline changes are an important component of Rayleigh interference data, there is no physical reason for allowing such time-varying offsets in the analysis of FDS-SV data, and usually excellent fits can are achieved with a constant baseline if the correct model is used.) The large misfit of this impostor model, especially in the initial boundaries, emphasizes the impact of the psFP signal change on the measured boundary amplitudes, not only the plateaus, and is therefore a very useful extension in particular of the discussion of the data from the rsEGFP2/FITC-BSA mixture. For the mixture including Padron a standard analysis fails even more clearly, affirming the previous point.

We also agree that the labels in the distribution plots were not very clear, and we have improved this in the revision.

*- The manuscript could be more easy to follow with some additional simple diagrams of the concepts or methodological design added to the figures. For example:a) small cartoon of the binding model and how the two signal components relate to this binding model in Figure 8) Figure 5: Were scans taken at regular intervals? If so would be useful to show on graph when the 2 min exposure happened. A diagram of the timing of events would be useful.c) Figure 1: hard to see different scans – perhaps show less scans? Also what are the purple scans? Are these run in parallel AUC cells. Also it is not clear what the sequence of events are in collecting this data. Perhaps a diagram showing the temporal series of events could be used to illustrate the concept more clearly next to the data.*

We thank the reviewers for encouraging the inclusion of more diagrams. We have followed this recommendation and feel they clarify the concepts considerably. With regard to the specific points:

a) We have included a cartoon of the binding scheme of GluA2-ATD and GluA3-ATD with the competitive homo- and hetero-dimerization as a new Figure 7. For greater clarity, the schematics introduces indices to better distinguish the different Kd-values.

b) We have included a schematics of the timing of events during the blinking experiments in Figure 5 (now Figure 6). We have also clarified in the Figure legend that the visual gap is mostly not from a prolonged time between scans (it is only extended by 50% to accommodate the 405 nm exposure) but from the significant change in the boundary signal magnitude. On first glance, one tends to compare non-equivalent points in the spreading boundary, but on closer inspection one can recognize the boundary midpoints are not as far apart.

c) In Figure 1 (now Figure 2), we have exchanged data in panels c and e to better display the scans. However, the individual data points are still hard to see. At the root of the problem is the fact that a lot of the scans cross, which does not usually happen in traditional SV.

To address this concern differently, we have added movies for all sedimentation processes shown in Figure 2 through Figure 2. They are referred to from within the figure legend, and allow very clear and unambiguous inspection of the signal evolution with time. The associated videos should be easy to retrieve from the *eLife* website directly or from links embedded in the article pdf.

Still, we feel it is useful to allow a visual comparison of the different types of temporal modulation in side-by-side overlay plots, since such overlay plots such as panel b are the traditional form of SV data presentation. We have experimented with various formats, and came to the conclusion that the extended rainbow color scheme from purple-blue-green-red works best. Reducing the number of scans does not always offer a significant improvement in overall clarity, since it has the disadvantage of making the temporal progression harder to follow. However, we have improved the resolution of the plots, which should allow a reader to zoom in when reading an electronic copy.

Finally, the temporal series of events should now be clearer with the new schematics in Figure 6, and the help of the movies.

*- Subsection “Application to GluA2 – GluA3 ATD heteromers”: the GluA3 data should be shown for completeness, perhaps added onto Figure 6.*

Thank you for this suggestion. The revised figure (now Figure 8) shows the requested rsEGFP2-GluA3 data.